# Household air pollution is associated with disease severity in Ugandan children hospitalized with hypoxemic pneumonia

Ayla Ahmed[1], Sehaj Sandhu[2], Sophie Namasopo[3], Juliet Nabwire[4], Qaasim Mian[2], Andrea L. Conroy[5], Jackson Amone[6], Charles Olaro[6], Robert O. Opoka[4,7], Michael T. Hawkes[8]*

1 Department of Medicine, University of St Andrews, St Andrews, United Kingdom, 2 Department of Pediatrics, University of Alberta, Edmonton, Alberta, Canada, 3 Department of Pediatrics, Kabale District Hospital, Kabale, Uganda, 4 Global Health Uganda, Kampala, Uganda, 5 Department of Pediatrics, Indiana University School of Medicine, Indianapolis, Indiana, United States of America, 6 Ministry of Health, Kampala, Uganda, 7 Department of Pediatrics, Medical College East Africa, Aga Khan University, Nairobi, Kenya, 8 Department of Pediatrics, University of British Columbia, Vancouver, Canada

* michael.hawkes@bcchr.ca

## Abstract

### Background

Household air pollution (HAP) due to biomass fuel use is a risk factor for childhood pneumonia, a leading cause of under-five mortality globally. This study examined the relationship between HAP and disease severity in Ugandan children hospitalized with hypoxemic pneumonia.

### Methods

We conducted a retrospective case-control study across 20 Ugandan hospitals. $PM_{2.5}$ exposure was estimated using caregiver-reported fuel use, cooking duration, kitchen structure, and ventilation. Pneumonia severity was assessed clinically, and associations were analyzed using non-parametric tests.

### Results

We included 735 children (median age 9 months, 42% female) hospitalized with hypoxemic pneumonia. Most households used firewood (84%) or charcoal (16%) for cooking. Other HAP sources included cigarette smoke (17%) and open-flame lighting (17%). The median estimated personal $PM_{2.5}$ exposure was 145 µg/m³ (IQR 79–270), and 732 children (99.6%) exceeded the recommended WHO limit of 15 µg/m³. Chronic HAP-related symptoms included cough (57%), red eyes (41%), rhinorrhea (38%), difficulty breathing (22%), and wheeze (6.5%). Higher PM2.5 exposure was significantly associated with more frequent red eyes (p = 0.0073), rhinorrhea (p = 0.016), and difficulty breathing (p < 0.0001). The median SICK score was 3.9

**Data availability statement:** All relevant data are within the paper and its Supporting information file.

**Funding:** This study was supported by Grand Challenges Canada (Grant Number 1909-27795 [MH]) and The Women and Children's Health Research Institute (Reference Number WCHSSLDRP 2371). The funders had no role in study design, data collection and analysis, decision to publish, or preparation of the manuscript.

**Competing interests:** The authors have declared that no competing interests exist.

(IQR 3.4–5.0). Higher scores correlated with higher $PM_{2.5}$ exposure ($\tau = 0.15$, $p < 0.0001$) and increased mortality risk ($p = 0.0024$). Higher $PM_{2.5}$ exposure was also linked to WHO danger signs, lower $SpO_2$ at admission, longer duration of oxygen therapy, and greater total oxygen volume administered over the hospital admission ($p < 0.05$ for all).

## Conclusion

Household $PM_{2.5}$ exposure from biomass combustion is associated with greater pneumonia severity in hospitalized children under five. Reducing household air pollution through cleaner fuels, better ventilation, and behavioral changes may improve outcomes for a leading cause of child mortality globally.

## Introduction

Beyond the neonatal period, the leading cause of mortality in children under five years of age is pneumonia, accounting for 13% of deaths world-wide [1]. Fifteen low-income countries account for three quarters of all the worldwide pneumonia episodes among children under 5, with the highest number of cases occurring in South Asia and sub-Saharan Africa [2].

Biomass combustion is the primary source of domiciliary energy for over 2 billion people worldwide. Approximately 75% of households in low- and middle-income countries depend on this form of fuel for cooking, heating, and lighting [3,4]. Biomass can include sources such as wood, coal, and cow dung. Burning of biomass fuels releases particulate matter (PM) and toxic pollutants such as sulphur dioxide, carbon monoxide (CO), and other organic compounds into the household environment [5]. PM may be quantified as the density of fine particles <2.5 μm size ($PM_{2.5}$) [6], which may have adverse health effects such as acute respiratory tract infection, increased airway hyperresponsiveness, and decreased lung growth [3,7]. By increasing the risk and severity of respiratory illness, household air pollution (HAP) accounts for over 4 million deaths annually [7].

In this study, we describe cooking practices, household biofuel use, and a summary estimate of HAP in a large, well characterized country-wide cohort of Ugandan children hospitalized with hypoxemic pneumonia. The primary objective was to examine the association between indoor pollution and disease severity. Findings from this study will inform public health interventions in the rural African context to address HAP as a modifiable risk factor for childhood pneumonia.

## Methods

### Study design

This was a retrospective case-control study examining the association between exposure to HAP and disease severity among children hospitalized with hypoxemic pneumonia. The study was a secondary analysis of the Solar Oxygen Study [8].

## Study participants and setting

Inclusion criteria for the parent trial (n = 2405) were: under five years of age; cough or difficulty breathing; peripheral $O_2$ saturation ($SpO_2$) <92%; and requiring hospitalization. Exclusion criteria were: known cyanotic congenital heart disease; or hypoxemic ischemic encephalopathy. For this secondary analysis, we further excluded patients who tested positive for malaria, patients for whom oxygen was not available, patients with $SpO_2 \geq 90\%$, and those who did not complete a questionnaire for HAP exposure.

Patients included in the parent study were hospitalized at one of the following facilities in Uganda: Gombe District Hospital (DH); Karisizo DH; Kayunga DH; Ssembabule Health Centre (HC) IV; Bugobero HC IV; Bukedea HC IV; Bumanya HC IV; Kidera HC IV; Muyembe HC IV; Kamuli DH; Adumi HC IV; Atiak HC IV; Lalogi HC IV; Kitgum DH; Apac DH; Bundibugyo DH; Kagadi DH; Kitagata DH; Kyenjojo DH; Lyantonde DH.

## Measurement of exposure to HAP

We estimated the personal exposure of a young child to HAP using a log-linear model linking the kitchen concentration of $PM_{2.5}$ to household variables (S1 Formula) [9]. This equation has been previously used for global burden of disease modeling [10], with correlation (r = 0.56) between predicted and measured values [9]. The model uses the following inputs to estimate the kitchen area $PM_{2.5}$: fuel type, kitchen type, kitchen ventilation, and cooking duration. An additional variable from the original study, representing the state in India, was not included in our calculation. The ratio between the daily average personal exposure and kitchen concentration (0.628 for young children) was applied to estimate exposures for children in our study [10]. Household variables were collected using a standardized questionnaire, administered to the caregiver accompanying the hospitalized child, in the language best understood.

## Measurement of outcome: pneumonia severity

As the primary measure of clinical severity, we used the Signs of Inflammation in Children that Kill (SICK) score, a composite index of disease severity, developed for use in resource-constrained settings. Higher SICK scores are associated with progressively increasing risk of mortality [11]. The SICK score was calculated upon initial hospital admission as previously described [12]. The score (range 0–8.6) was computed as the weighted sum of the following clinical variables: age < 1 month (+2.2) or < 12 months (+1.0) or < 5 years (+0.3); temperature > 38°C or < 36°C (+1.2); heart rate > 160 minutes$^{-1}$ for infants or > 150 minutes$^{-1}$ for children (+0.2); respiratory rate > 60 minutes$^{-1}$ for infants or > 50 minutes$^{-1}$ for children (+0.4); systolic blood pressure < 65 mmHg for infants or < 75 mmHg for children (+1.2); oxygen saturation < 90% (+1.4); capillary refill time > 3 seconds (+1.2); and level of consciousness less than "alert" (+2.0).

## Sample size calculation

In our primary analysis, we tested the hypothesis that household air pollution ($PM_{2.5}$, continuous variable) would be positively correlated with disease severity (SICK score, continuous variable). We estimated that we would need a sample size of 618 to detect a correlation of 0.1 or more, with 80% power at the $\alpha = 0.05$ (one-sided) level of significance (package *pwrss* [13] in the R statistical environment). The formula for the sample size calculation is provided in the Supporting information (S2 Formula).

## Statistical analysis

Descriptive statistics used median and interquartile range (IQR) for continuous variables and number with percentage for binary or categorical variables. For comparative statistics, the non-parametric Mann–Whitney *U* test was used for continuous data. The two-tailed Pearson $\chi^2$ or Fisher exact test was used for categorical data, as appropriate. Correlations between continuous variables used the non-parametric rank correlation coefficient (Kendall's tau-B, τ). To quantify the

association between the $PM_{2.5}$ and the SICK score, we used a linear regression model. The dependent variable was the SICK score and the independent variable was the $\log_{10}(PM_{2.5})$. Logarithmic transformation was used because the $PM_{2.5}$ was right-skewed and the measurement error scales proportionally with the magnitude of the concentration. The odds ratio (OR) and its 95% confidence interval (CI) were used to quantify the degree of association between binary variables. The OR was calculated as the cross-product of the 2×2 contingency table $\left(OR = \frac{ad}{bc}\right)$ and the confidence interval was calculated using the maximum unconditional likelihood (Wald) method (normal approximation on the logarithmic scale) with the standard error calculated as $SE_{\ln(OR)} = \sqrt{\frac{1}{a} + \frac{1}{b} + \frac{1}{c} + \frac{1}{d}}$. Non-parametric statistics were used to avoid the assumption of normally distributed data. GraphPad Prism version 6 (GraphPad Software Inc., La Jolla, CA, USA, 2012) and R (version 4.3.0) were used for data analysis and visualization.

## Ethical approval

The study was reviewed and approved by the Makerere University School of Biomedical Sciences Research Ethics Committee (SBSREC-644), Uganda National Science and Technology (HS 2569), and the University of Alberta Health Research Ethics Board (Reference Pro00084784).

Parents or legal guardians of children (all under five years of age) provided written informed consent for study participation.

The dates when data were accessed for research purposes (patient recruitment and data collection) were: 01/07/2019 to 30/11/2021.

## Results

We included 735 children hospitalized with hypoxemic pneumonia between 1 July 2019 and 30 Nov 2021 (Fig 1). Clinical characteristics were summarized (Table 1). Of note, male children were over-represented in the cohort (male to female ratio 1.4:1, p<0.0001).

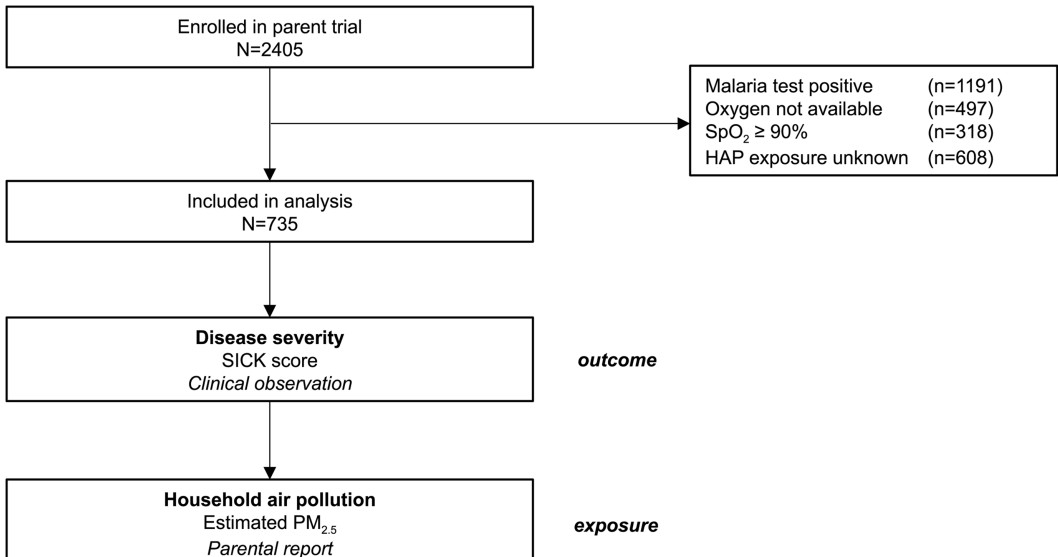

**Fig 1. Trial flow diagram.** The parent trial was the Solar Oxygen Study (N=2405) [8], which enrolled children under 5 years of age hospitalized with oxygen saturation ($SpO_2$) < 92%. Malaria-negative patients with $SpO_2$<90% who received supplemental oxygen and who completed the questionnaire for household air pollution (HAP) were included in the current secondary analysis (N=735). The outcome was disease severity, measured using the composite clinical score, Signs of Inflammation in Children that Kill (SICK), and assessed by the treating clinician. The exposure was household air pollution, quantified using a HAP score [9], and assessed retrospectively by parental report of household and cooking fuel characteristics.

**Table 1. Characteristics at admission of 735 hospitalized children.**

| | Hypoxemic pneumonia (N = 735) |
|---|---|
| **Demographics** | |
| Age [mo], median (IQR) | 9 (3-18) |
| Female sex | 309 (42) |
| Multidimensional poverty index poor [a] | 397 (54) |
| **Clinical features (parental report)** | |
| Tactile fever | 606 (82) |
| Cough | 706 (96) |
| Difficulty breathing | 703 (96) |
| Convulsions | 64 (8.7) |
| Altered consciousness | 164 (22) |
| Vomiting everything | 157 (21) |
| Unable to feed/drink | 396 (54) |
| **Physical exam** | |
| Severely underweight [b] | 81 (11) |
| Oxygen saturation [%], median (IQR) | 84 (79-87) |
| Tachypnea [c] | 439 (60) |
| Tachycardia [c] | 255 (35) |
| Temperature [°C], median (IQR) | 38 (37-38) |
| Level of consciousness | |
| Alert | 559 (76) |
| Response to voice | 68 (9.3) |
| Response to pain | 83 (11) |
| Unresponsive | 24 (3.3) |
| Deep breathing | 322 (44) |
| Chest indrawing | 704 (96) |
| Grunting | 451 (61) |
| Stridor | 281 (38) |
| Cyanosis | 112 (15) |
| WHO danger signs [d] | 605 (82) |
| **Composite index of disease severity** | |
| SICK, median (IQR) | 3.9 (3.4-5.0) |

Numbers represent n (%) unless otherwise stated.

IQR, interquartile range; RISC, respiratory index of severity in children; WHO, World Health Organization; SICK, Signs of Inflammation in Children that can Kill [11].

[a]The multidimensional poverty index (MPI) was calculated according to the method of Alkire and Foster [14]. A value >1/3 was considered MPI-poor.

[b]Weight-for-age below −3SD [15].

[c]Vital sign >99 percentile for age [16].

[d]WHO danger signs include convulsions, altered consciousness, vomiting everything, stridor at rest, and being unable to feed or drink.

We summarized household characteristics related to HAP (Table 2). The majority (84%) of patients used firewood as cooking biofuel. Charcoal was the second most common cooking fuel (16%). Other, cleaner, fuel sources (e.g., gas or electrical cookers) were used by less than 1% of households. The kitchen was outdoors in 33% of homes, in a structure

separate from the sleeping area (main house) in 63%, and inside the main house in 3.6%. The household stove configurations were categorized as follows: three-stone fires (n = 581, 79%), traditional charcoal jikos (n = 114,16%), and fixed chimney stoves (n = 30, 4.1%). Cooking fire burned for a median of 6 hours per day (IQR 4–8), including preparation time and the time that the fire burned afterwards. Both preparation time (p = 0.016) and post-cooking burn time (p = 0.0017) were longer in households using wood compared to charcoal fuel. The number of household members partaking in daily meals was median 5 (IQR 4–7) and was correlated with preparation time (τ = 0.061, p = 0.031). The cost of the cooking fuel was associated with fuel type: 496/540 (92%) of firewood-burning households obtained the fuel for free, compared to 13/96 (14%) of charcoal-burning households (p < 0.0001). The daily cost of charcoal was median USD$0.54 (IQR 0.27–0.81).

Exposure to HAP sources other than cooking biofuels included household cigarette smoking (16%) and lighting (e.g., open wick lamp, 17%, and hurricane lamp, 11%). A large number of households had access to a clean source of lighting, solar powered lamps (49%).

HAP exposure was estimated based on self-reported household characteristics. The estimated personal exposure to fine particulate matter ($PM_{2.5}$) was median 145 µg/m$^3$ (IQR 79–270). There were 732 children (99.6%) with daily exposure exceeding 15 µg/m$^3$, the limit recommended by WHO guidelines [17] and 484 (66%) exceeded 100 µg/m$^3$, a level previously associated with higher risk of pneumonia [18].

Parental report of symptoms related to smoke exposure included cough (57%), red eyes (43%), rhinorrhea (35%), difficulty breathing (21%) and wheeze (6.9%) (Table 2). Wood fuel was associated with 2.0-fold higher odds (95% CI 1.2–3.4) of red eyes relative to charcoal (p < 0.0001). Higher $PM_{2.5}$ exposure was associated with a higher frequency of red eyes, difficulty breathing, and runny nose (Fig 2).

The median composite disease severity score (SICK) was 3.9 (IQR 3.4 to 5.0). There was a statistically significant correlation between the $PM_{2.5}$ and the SICK score (τ = 0.15, p < 0.0001, Fig 3A). This corresponds to a 0.52 point (95%CI 0.33–0.70, p < 0.0001) increase in the SICK score for every 10-fold increase in the $PM_{2.5}$. In turn, higher SICK score was associated with higher risk of mortality (p = 0.0024, Fig 3B). Higher $PM_{2.5}$ exposure was also associated with WHO danger signs (Fig 3C) and compromised oxygenation, as evidenced by lower $SpO_2$ at admission, longer duration of oxygen therapy, and larger total volume of supplemental O2 administered (Fig 3D-F).

## Discussion

Here we show an association between HAP and disease severity in a country-wide cohort of Ugandan children under five years of age hospitalized with hypoxemic pneumonia. We found a statistically significant correlation between $PM_{2.5}$ exposure and the SICK score (τ = 0.15, p < 0.0001), which, in turn, was linked to a higher risk of mortality. Our study confirms and extends previous findings from other LMICs. Previous studies have explored the association between HAP and childhood pneumonia, using a variety of study designs and methods to measure both exposure (HAP) and outcome (pneumonia incidence or severity) (Table 3). A systematic review and meta-analysis of published studies concluded that the odds of childhood pneumonia are approximately 1.8 times higher in households using unprocessed solid fuels than those using cleaner fuels (e.g., liquefied petroleum gas, kerosene, or electricity) [19]. Our study is noteworthy for its large sample size, its focus on hospitalized patients with severe pneumonia, and its validated method of estimating $PM_{2.5}$.

We estimated the median daily personal exposure to HAP ($PM_{2.5}$) to be 145 µg/m$^3$. This is higher than a recent multi-centre study (e.g., median household exposure 48 µg/m$^3$ in Blantyre, Malawi) [47] but similar to a 2013 study from Bangladesh, in which household $PM_{2.5}$ concentrations exceeded 100 µg/m3 for a median of 5.3 hours per day [18]. We utilized a published method of estimating $PM_{2.5}$ levels based on self-reported data, such as cooking fuel and kitchen type [9]. While many previous studies have similarly employed questionnaires asking about fuel use [18,20–25,27–30,32,40,41], whether the child is carried on the mother's back during cooking [33–37], or general smoke exposure [31,38,39], others have attempted direct measurements of HAP by quantifying specific air pollutants, including CO [43,45,46,48], $PM_{2.5}$ [18,22], $NO_x$ [43], and $PM_{10}$ [41,42,48,49]. Measurement methods included $PM_{2.5}$ air samplers [18,22], $PM_{10}$ air samplers

**Table 2. Exposure to household air pollution among 735 children hospitalized with hypoxemic pneumonia.**

| | Hypoxemic pneumonia (N = 735) |
|---|---|
| ***Cooking practices*** | |
| Cooking fuel | |
| Firewood | 615 (84) |
| Charcoal | 116 (16) |
| Other | 4 (0.54) |
| Type of stove | |
| Stones (open fire) | 581 (80) |
| Charcoal stove | 114 (16) |
| Fireplace with chimney | 30 (4.1) |
| Other | 4 (0.55) |
| Timing of cooking | |
| Morning | 641 (88) |
| Afternoon | 527 (72) |
| Evening | 476 (65) |
| Night | 184 (25) |
| Cooking for how many people? | 5 (4-7) |
| Cooking time [h], median (IQR) | 4 (3-6) |
| Time fire burns after cooking [h], median (IQR) | 0.5 (0.17-2) |
| Total time fire burning [h], median (IQR) | 6 (4-8) |
| ***Kitchen and ventilation*** | |
| Kitchen location | |
| Separate structure | 464 (63) |
| Outdoor kitchen | 241 (33) |
| Indoor kitchen | 26 (3.6) |
| Windows in bedroom | 629 (86) |
| Additional ventilation in bedroom | 606 (83) |
| Windows in kitchen | 282 (38) |
| Additional ventilation in kitchen | 437 (59) |
| ***Other sources of HAP*** | |
| Smoker | 124 (17) |
| Lighting | |
| Solar powered | 362 (49) |
| Battery | 41 (5.6) |
| Open wick lamp | 124 (17) |
| Hurricane lamp | 83 (11) |
| Other | 122 (17) |
| ***Symptoms around smoke*** | |
| Cough | 239 (57) |
| Red eyes | 182 (43) |
| Rhinorrhea | 149 (35) |
| Difficulty breathing | 87 (21) |
| Wheeze | 29 (6.9) |
| ***Composite index of exposure*** | |
| Estimated $PM_{2.5}$ [μg/m³] [a], median (IQR) | 140 (79-270) |

[a]Personal exposure to fine particulate matter (< 2.5 μm), estimated from questionnaire data.

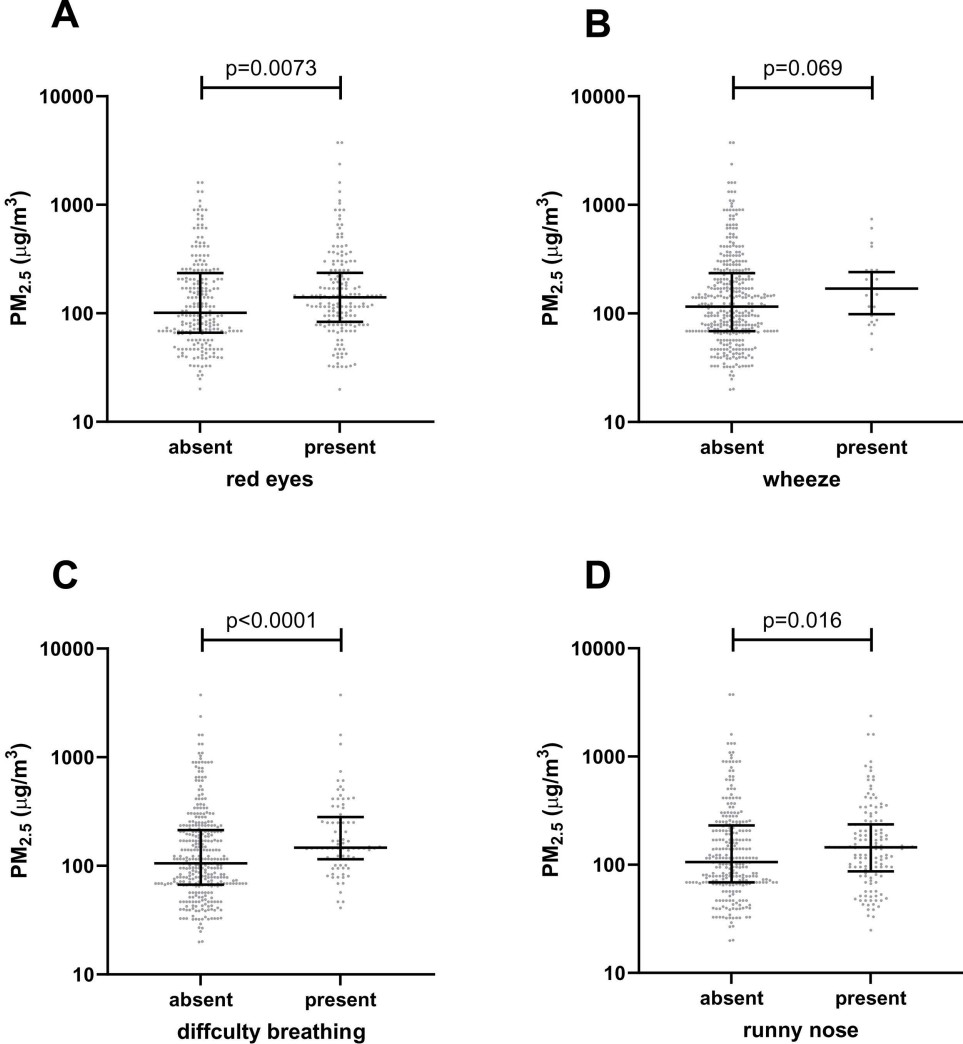

**Fig 2. Association between parental report of chronic respiratory tract symptoms and estimated household air pollution (PM$_{2.5}$).** A. Red eyes; **B**. wheeze; **C**. difficulty breathing; and **D**. runny nose.

[41,42,48,49], Lascar monitors [46], LD-3K fine dust monitors [43], and NO$_2$ badges [43]. These approaches provide direct and objective measurements but are often complex, costly, and can be influenced by spatial and temporal variability in pollutant levels within and outside the household. While our study lacks direct HAP measurements and instead relies on self-reported survey data, it overcomes the limitations of simpler dichotomous exposure questionnaires (e.g., biofuels for cooking) by utilizing a validated log-linear model to estimate PM$_{2.5}$ exposure [9]. Thus, our exposure measurement was quantitative yet practical. This may be a pragmatic, feasible, reproducible model for future studies of HAP in low-resource settings.

We examined the severity of pneumonia among hospitalized children as our primary outcome of interest, using the composite clinical severity score, SICK [11]. Past studies have used alternative outcomes relevant to childhood pneumonia, including caregiver-reported symptoms such as rapid or difficult breathing and cough [18,20–22,24,25,29,34,45,50], fieldworker observations of ARI symptoms during home visits [18,32,33,38,42,46,50], and diagnoses made by physicians

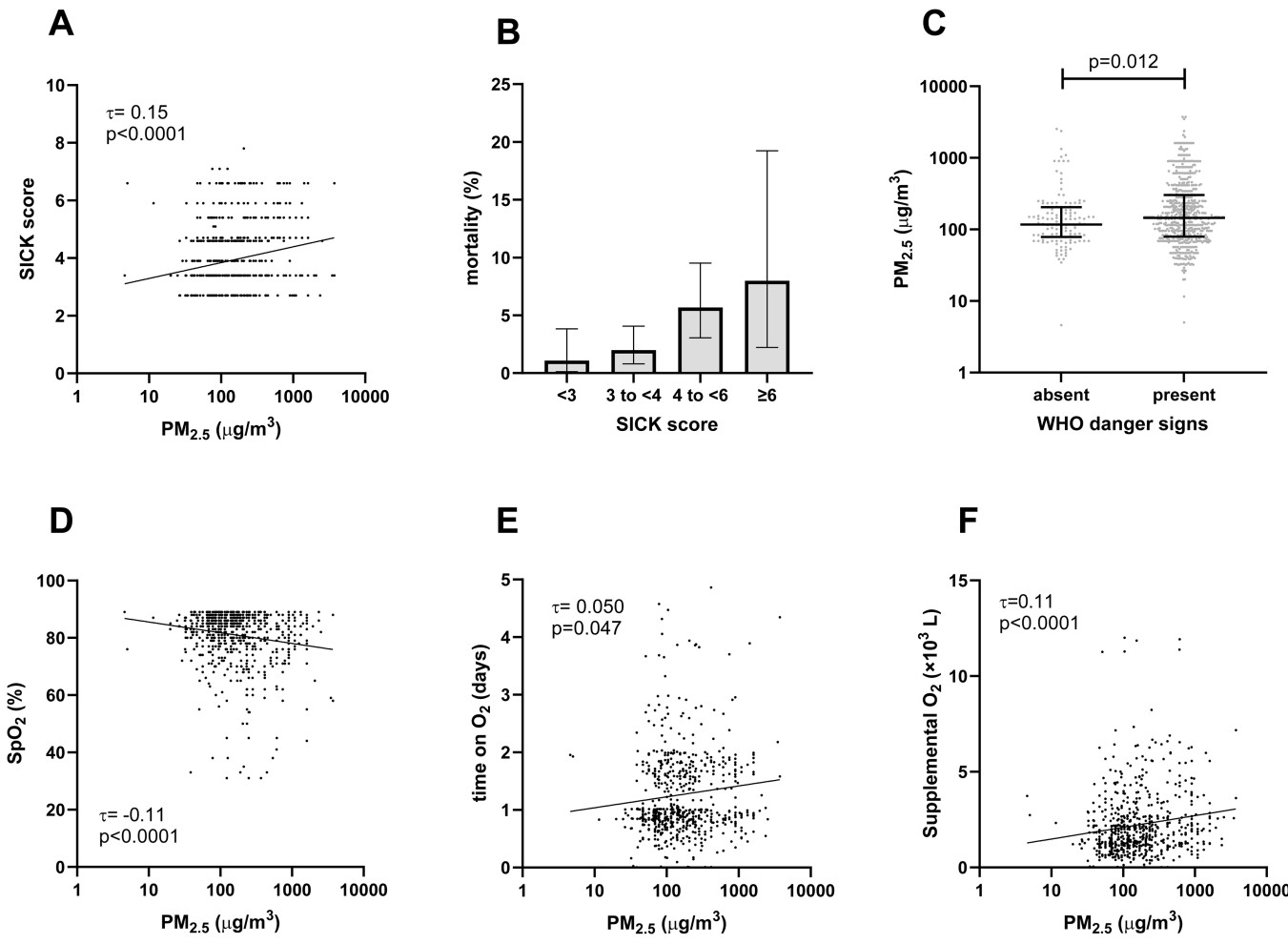

**Fig 3. Association between disease severity and estimated daily average personal household air pollution exposure (PM$_{2.5}$).** A. There was a statistically significant correlation between the PM$_{2.5}$ and composite severity score (Signs of Inflammation in Children that Kill, SICK, Kendall's tau-B, $\tau=0.15$, $p<0.0001$). B. Higher SICK score was associated with mortality. C. Higher PM$_{2.5}$ was seen in children with WHO danger signs. **D-F**. The peripheral oxygen saturation (SpO$_2$) was lower (D), the duration of oxygen therapy was longer (E), and the total volume of oxygen administered was higher (F) in patients exposed to higher levels of household air pollution.

[18,28,30,35,36,39,40,44,45,50], though standardized protocols were not always reported. Other studies have used more objective definitions of pneumonia, including radiological findings [33,50], assessments of oxygen saturation [23,50], or laboratory tests for respiratory syncytial virus (RSV) [23,31,37] or pneumococcus [35]. Some studies were community-based, in which most pneumonia episodes were likely mild [21,22,27,29,33]. Fewer studies examined hospitalized patients [28,31,36,37,39,41,44], severe pneumonia episodes [36,40,44,46], and fatal cases [26,34,50]. Our study is noteworthy for its inclusion of hospitalized, hypoxemic patients at high risk of mortality. Pneumonia severity was measured objectively, quantitatively, and in a standardized manner using the validated SICK clinical score. While past studies have demonstrated the association between HAP and incident pneumonia, our study links HAP to pneumonia severity in a cohort at high risk of mortality.

Supporting the deleterious effects of HAP, symptoms of airway and mucous membrane inflammation were common in our study. These observations are consistent with previous findings, where children exposed to biofuel smoke exhibited

**Table 3. Association between household air pollution (HAP) and childhood pneumonia in previous studies.**

| First Author | Year | Country | Design | Number of Patients | Exposure (HAP) | Outcome | Magnitude of Association | Ref |
|---|---|---|---|---|---|---|---|---|
| a. Studies that asked the caregiver about the type of household cooking fuel | | | | | | | | |
| Mishra | 2006 | India | Cross-sectional (survey) | 29,768 children (0–35 months) | Questions on fuel type(s) mainly used for cooking: biofuels | ARI during the 2 weeks before interview (parental survey) | OR = 1.82 (95% CI 1.58–2.09) | [20] |
| Naz | 2020 | Pakistan | Cross-sectional (3 surveys) | 28,919 children U5 | Questions on fuel type(s) mainly used for cooking: polluting fuel | Childhood pneumonia based on maternal report of symptoms | OR = 1.25 (95% CI 1.05–1.35) | [21] |
| Sanbata | 2014 | Ethiopia | Cross-sectional | 422 house-holds, children U5 | Questions on fuel type(s) mainly used for cooking: solid biomass fuel | ARI, based on maternal report of symptoms | aOR = 2.96 (95% CI 1.38–3.87) | [22] |
| Al-Sonboli | 2006 | Yemen | Case-control | 601 children <2 years old hospitalized | Questions on fuel type(s) mainly used for cooking: non-gas cooking fuels | ARI with hypoxemia ($SpO_2$ <88%) and RSV (RT-PCR) | OR = 10.3 (95% CI 2.2–48) | [23] |
| Islam | 2022 | Bangla-desh | Cross-sectional (survey) | 8321 children U5 | Questions on fuel type(s) mainly used for cooking: solid fuels | ARI, based on caregiver report of symptoms | OR = 1.69 (95% CI 1.05–2.72) | [24] |
| Wichmann | 2006 | South Africa | Cross-sectional (survey) | 4679 children U5 | Questions on fuel type(s) mainly used for cooking: polluting fuels | ARI, based on maternal report of symptoms | OR = 1.27 (95% CI 1.05–1.55) | [25] |
| Bassani | 2010 | India | Case-control | 616, 391 chil-dren U5 | Fieldworker observation of fuel type used for cooking: solid fuel use | 1. Non-fatal pneumonia 2. Child deaths in the previous year (household survey) | 1. PR = 1.54 (boys) PR = 1.94 (girls) 2. PR = 1.30 (boys) PR = 1.33 (girls) | [26] |
| Dhimal | 2010 | Nepal | Cross-sectional | 41 313 children U5 | Questions on fuel type(s) mainly used for cooking: solid biomass fuel | ARI and pneumonia, based on caregiver recall and treatment records. | AF = 0.496 | [27] |
| Mahalanabis | 2002 | India | Case-control | 262 children aged 2–25 months | Questions on fuel type(s) mainly used for cooking: solid fuel | Physician-diagnosed pneumonia | OR = 3.97 (95% CI 2.00–7.88) | [28] |
| Budhathoki | 2020 | Nepal | Cross-sectional (3 surveys) | 15, 372 chil-dren U5 | Questions on fuel type(s) mainly used for cooking: polluting fuels | Childhood pneumonia, based on symptoms reported by parents | aRR = 1.98 (95% CI 1.01–3.92) | [29] |
| Bhat | 2008 | India | Case-control | 214 children U5 | Questions on fuel type(s) mainly used for cooking: fuel other than LPG | Clinically identified ALRI using WHO case definitions | OR = 4.63 (95% CI 1.67–12.97) | [30] |
| Jeena | 2003 | South Africa | Prospective observa-tional | 114 children aged 3–24 months | Questions on fuel type(s) mainly used for cooking: polluting fuel | RSV-confirmed bronchiolitis hospital admissions | Higher admission risk with noxious fuels (p = 0.05). | [31] |
| Sharma | 1998 | India | Prospective cohort | 642 infants | Questions on fuel type(s) mainly used for cooking: wood or kerosene | Fieldworker-observed ALRI episodes during twice-weekly house visits | Kerosene: 31.7% ALRI Wood: 19.0% ALRI | [32] |
| b. Studies that asked the caregiver about proximity to cooking fire | | | | | | | | |
| Armstrong | 1991 | The Gambia | Prospective cohort | ~500 children U5 | Questions on whether the child is carried by the mother during cooking | ALRI episodes iden-tified via weekly field visits and chest X-rays | OR = 1.9 (95% CI 1.0–3.9) | [33] |
| De Francisco | 1993 | The Gambia | Case-control | 129 children <2 years old | Questions on whether the child is carried by the mother during cooking | ALRI deaths identified via verbal autopsy | aOR = 5.2 (95% CI 1.7–16) | [34] |

*(Continued)*

**Table 3.** (Continued)

| First Author | Year | Country | Design | Number of Patients | Exposure (HAP) | Outcome | Magnitude of Association | Ref |
|---|---|---|---|---|---|---|---|---|
| O'Dempsey | 1996 | The Gambia | Case-control | 239 children U5 | Questions on whether the child is carried by the mother during cooking | Physician and micro-biologic diagnosis of pneumococcal | OR = 2.55 (95% CI 0.98–6.65) | [35] |
| Howie | 2016 | The Gambia | Case-control | 1581 children aged 2–59 months | Questions on whether the child is carried by the mother during cooking | Severe vs. non-severe pneumonia, based on presenting symptoms | aOR = 1.7 (95% CI 1.0–3.0) | [36] |
| Weber | 1999 | The Gambia | Case-control | 641 children U5 | Questions on whether the mother cooks at least once daily | Physician-diagnosed ALRI due to RSV | aOR = 0.31 (95% CI 0.14–0.70) | [37] |
| Pandey | 1989 | Nepal | Prospective observational | Eligible children < 2 years old | Questions on time spent near the fireplace per day >2 hrs/day near fireplace | Fieldworker-observed ARI episodes, graded by severity (Grades I-IV) | linked to higher ARI risk | [38] |
| Karki | 2014 | Nepal | Case-control | 200 children U5 | Questions regarding household use of a smoky cook-stove located indoors | Physician-diagnosed pneumonia in hospitalized children | OR = 3.76 (95% CI 1.20–11.82) | [39] |
| PrayGod | 2016 | Tanzania | Case-control | 117 children (2–59 months) | Questions on location of kitchen: indoor cooking | Physician-diagnosed severe or very severe pneumonia | OR = 5.5 (CI 1.4–22.1) | [40] |
| **c. Studies that measured HAP directly** | | | | | | | | |
| Robin | 1996 | United States of America | Case-control | 90 children aged 1–24 months | Domestic $PM_{10}$ levels ≥ 65 μg/m³ (air-sampling pump) | Hospitalization for ALRI before interview (parental survey) | OR = 7.0 (95% CI 0.9–56.9) | [41] |
| Ezzati | 2001 | Kenya | Prospective cohort | 94 children aged 0–4 years | Domestic $PM_{10}$ levels >3500 μg/m³ (personal and stationary monitoring devices) | ARI, based on fieldworker observation of symptoms | OR = 6.73 (95% CI 3.75–12.06) | [42] |
| Kilabuko | 2007 | Tanzania | Cross-sectional | 100 households | Domestic $PM_{10}$, CO, and $NO_2$ levels measured from biofuel cooking smoke | ARI diagnosed based on questionnaire answered by the household's cook | OR = 5.5 (95% CI 3.6–8.5) | [43] |
| Gurley | 2013 | Bangladesh | Prospective cohort | 257 children (followed 0–24 months) | Domestic $PM_{2.5}$ levels >100 μg/m³ (PM air monitor) | Physician-diagnosed ALRI | aIRR = 1.07 (95% CI 1.01-1.14) per hour | [18] |
| **d. Study on smoking exposure** | | | | | | | | |
| Shah | 1994 | India | Case-control | 400 children U5 | Questions on behavioral habits: smoking in household | Physician-diagnosed severe ARI | OR = 1.15 (95% CI 0.72–1.84) | [44] |
| **e. Randomized controlled trials in which the intervention aimed to reduce HAP** | | | | | | | | |
| Smith | 2011 | Guatemala | RCT | 534 households | Intervention to reduce HAP: woodstove with chimney | Physician-diagnosed pneumonia | RaR = 0.84 (95% CI 0.63–1.13) | [45] |
| Kinney | 2021 | Ghana | RCT | 1141 infants | Intervention: clean cookstoves | Physician-diagnosed pneumonia and severe pneumonia | RR = 1.06 (95% CI 0.99–1,13) | [46] |

ARI; acute respiratory infection; ALRI; acute lower respiratory infection; aRR, adjusted relative risk; OR, odds ratio; U5, under five years of age; PR, prevalence ratio; LPG, liquid petroleum gas; AF, attributable fraction; RaR, rate ratio; aIRR, adjusted incidence rate ratio; RCT, randomized controlled trial.

similar symptoms [51,52]. Such effects are biologically plausible because air pollutants can cause inflammation of the airways and alveoli, increasing the severity of respiratory infections [53]. Moreover, children under five may be more susceptible to the effects of HAP than adults for several reasons: shorter stature with higher exposure to dense smoke; immature immune defense mechanisms [53]; narrower airways, which result in greater proportional airway obstruction; and a larger lung surface area per kilogram of body weight, along with a higher oxygen consumption rate, leading to increased inhalation of polluted air [54]. We specifically found that higher $PM_{2.5}$ concentrations are associated with an increased prevalence of rhinorrhea, difficulty breathing, and red eyes. Consistent with these findings, others observed a 39% increase in hourly cough rate in response to a 10-fold rise in hourly $PM_{2.5}$ concentrations [55].

Our study provided rich data on household cooking practices in rural Uganda, which may influence childhood exposure to HAP. Firewood remains the predominant cooking fuel, used by 84% of households. This reliance on firewood combustion may contribute to higher $PM_{2.5}$ exposure levels due to greater emissions compared to charcoal. For instance, charcoal emits lower $PM_{2.5}$ levels, averaging $0.21 \pm 0.06$ ppm [56], and has a lower $PM_{2.5}$ to $PM_{10}$ ratio than firewood [57]. Yet, despite being less polluting and offering shorter cooking times due to its higher burning efficiency (28% as opposed to 17%), charcoal adoption remains limited [58]. Its higher cost makes it less accessible than firewood, which is often freely available in rural areas. Consistent with our findings, firewood is often left burning long after cooking is completed, as seen in Kenyan households where cooking fires were reported to burn for 5–12 hours daily, further exacerbating $PM_{2.5}$ exposure [59]. Since firewood does not incur a monetary cost, families may allow it to continue burning even after cooking is complete, which may explain the longer cooking durations compared to charcoal. However, cooking with charcoal is far from an ideal alternative, and innovations in cleaner cooking technologies suitable for low-income settings are actively being explored. Households using cleaner fuel alternatives such as liquefied petroleum gas [60], ethanol cookstoves [61], and solar-powered cookstoves [62] represent important advances toward reducing HAP exposure. In parallel, chimney-equipped biomass stoves have shown potential to reduce HAP, with previous findings demonstrating an approximately 90% reduction in 48-hour kitchen CO concentrations compared to open wood fires [45]. Hence, these alternative cooking methods represent promising interventions for reducing pneumonia-related childhood morbidity and mortality, especially in rural communities. In addition, cost-effective behavioural change interventions (typically delivered through community counseling), such as promoting outdoor cooking, increasing ventilation while cooking indoors, and minimizing children's exposure to traditional indoor cooking fires, have also been effective in reducing household $PM_{10}$ and CO concentrations [63].

Our study has several limitations. This was a secondary analysis of data originally collected for a cluster randomized controlled trial [8]. We retrospectively assessed the exposure through parental reports of household characteristics and cooking practices. Thus, it may be subject to recall bias, limiting our ability to make causal inferences. Furthermore, we used modeled $PM_{2.5}$ exposure estimates based on household variables and calculated an estimated $PM_{2.5}$ exposure. While this estimate has a moderate correlation with measured values of $PM_{2.5}$, it remains a superior alternative to simple dichotomous fuel-type questionnaires, which are the standard in much of the existing literature [9]. This may have reduced the precision of the exposure assessment. Although our sample size was larger than many past publications, a study with more fatal cases would be necessary to show a direct association between HAP and mortality. Nonetheless, we demonstrated a plausible relationship between HAP and disease severity, and between disease severity and death. Our multi-center study included 20 Ugandan health facilities, which are likely representative of the country's public hospitals; however, the population was uniformly rural, low-income, and highly exposed to HAP through firewood cooking smoke. Results should therefore be extrapolated with caution to other countries, higher-income settings, and areas with lower use of cooking biofuels.

In summary, HAP appears to be a modifiable risk factor for severe childhood pneumonia, a leading cause of mortality among children under five globally. Our findings support initiatives aimed at reducing HAP as a strategy toward improved global child survival [64].

## Supporting information

**S1 Formula. Equation to predict household concentrations of particulate matter <2.5 µm in size (PM$_{2.5}$).**
(DOCX)

**S2 Formula. Equation to calculate the sample size for a Pearson correlation coefficient.**
(DOCX)

## Author contributions

**Conceptualization:** Michael T. Hawkes.

**Formal analysis:** Michael T. Hawkes.

**Methodology:** Michael T. Hawkes.

**Project administration:** Sophie Namasopo, Juliet Nabwire, Qaasim Mian, Robert O. Opoka.

**Supervision:** Sophie Namasopo, Robert O. Opoka.

**Writing – original draft:** Ayla Ahmed, Sehaj Sandhu.

**Writing – review & editing:** Sophie Namasopo, Juliet Nabwire, Qaasim Mian, Andrea L. Conroy, Jackson Amone, Charles Olaro, Robert O. Opoka, Michael T. Hawkes.

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
