## [Decision Letter · Decision Letter 0]

18 Mar 2026

PONE-D-25-36668Household air pollution is associated with disease severity in Ugandan children hospitalized with hypoxemic pneumoniaPLOS One

Dear Dr. Hawkes,

Thank you for submitting your manuscript to PLOS ONE. Though the paper has been reviewed by several reviewers, the responses of the comments are still missing. Please provide the responses to the comments and re-submit the revised paper according the comments made by all the reviewers and indicating how and where the comments have been addressed. Additionally, you are requested address some fresh comments made by me as mentioned below. Therefore, we invite you to submit a revised version of the manuscript that addresses the points raised during the review process. Academic Editor comments::1. The authors needs to provide the function from which the PM2.5 exposures were made from household characteristics along with its vital information about the model used..2.  Authors need to express the computational formula for the calculation of the sample size with input variable values. Only mentioning about the software is not appropriate.3, It needs to be clarified why non-parametric test was used instead of parametric test for measuring comparative statistics.4. It needs to be clarified how the odds ratios of different outcomes for risk factors were calculated along with their confidence intervals in the methodology section.5. I could not find the questionnaire for the SICK score calculation with reference. Please add it in supplementary.

We look forward to receiving your revised manuscript.

Kind regards,

Srijan Lal Shrestha, Ph.D.

Academic Editor

PLOS One

Journal Requirements:

2.Thank you for stating the following financial disclosure:  [This study was supported by Grand Challenges Canada (Grant Number 1909-27795 [MH]) and The Women and Children’s Health Research Institute (Reference Number WCHSSLDRP 2371).].

3. We note that there is identifying data in the Supporting Information file < Ahmed HAP data for analysis.csv>. Due to the inclusion of these potentially identifying data, we have removed this file from your file inventory. Prior to sharing human research participant data, authors should consult with an ethics committee to ensure data are shared in accordance with participant consent and all applicable local laws.

-Location data

Please remove or anonymize all personal information (ID, AGE,), ensure that the data shared are in accordance with participant consent, and re-upload a fully anonymized data set. Please note that spreadsheet columns with personal information must be removed and not hidden as all hidden columns will appear in the published file.

Additional Editor Comments:

The authors are requested to provide responses and revisions according to all the reviewers comments. Please kindly clarify how and where the comments have been addressed in the revised paper. Additional comments will be made only after going through the revised paper.

Reviewers' comments:

Reviewer's Responses to Questions

**Comments to the Author**

1. Is the manuscript technically sound, and do the data support the conclusions?

Reviewer #1: Yes

Reviewer #2: Yes

Reviewer #3: Yes

2. Has the statistical analysis been performed appropriately and rigorously? 

Reviewer #1: Yes

Reviewer #2: Yes

Reviewer #3: Yes

3. Have the authors made all data underlying the findings in their manuscript fully available?

Reviewer #1: Yes

Reviewer #2: Yes

Reviewer #3: Yes

4. Is the manuscript presented in an intelligible fashion and written in standard English?

Reviewer #1: Yes

Reviewer #2: Yes

Reviewer #3: Yes

5. Review Comments to the Author

Reviewer #1: This manuscript presents relevant information regarding the exposure to household (HAP) - and not traffic-related - air pollution and the sickness severity among Ugandan children. It is very impressive because Uganda is one of the many low- and middle-income countries that rely on biomass combustion for cooking, heating and lighting. This manuscript presents data from a noteworthy large sample size, 735 children from 20 hospitals, and convincing results and a very clear discussion. However, I still have some doubts on how they estimated the PM2.5 levels.

Here are some minor points that should be addressed before accepting this manuscript for publication.

Minor comments:

# We need more details about the PM2.5 data. Was it measured in µg or estimated, like an index? Describe it better.

-Lines 96-97: “We estimated the personal exposure of a young child to HAP using a log-linear model linking the kitchen concentration of PM2.5 to household variables [9].” Did you measure the PM2.5 concentration in the kitchens to estimate the real exposure? Mean measured PM2.5 should be presented in Table 2 too.

-Lines 101-103: “The ratio between the daily average personal exposure and kitchen concentration (0.628 for young children) was applied to estimate exposures for children in our study [10].” What exactly does this ratio mean? Explain better.

-“Statistically significant correlation between the PM2.5 and the SICK score (τ=0.15, p<0.0001). I cannot see this correlation in the graphic. What kind of variable was PM2.5 considered?

# Lines 282-284: “This estimate has a moderate correlation (r = 0.56) with measured values of PM2.5 [9]. This may have reduced the precision of the exposure assessment.” The R value of the correlation was already stated before, in the methods. So, you do not need to repeat it.

Reviewer #2: A. General Comments

This is a well-researched, logical, and highly consistent manuscript that addresses a critical gap in global health literature. While there is extensive literature linking household air pollution (HAP) to the incidence of pneumonia, this study provides significant additional value by focusing specifically on the severity and prognosis of pneumonia among a high-risk, hospitalized, hypoxemic cohort.

The large sample size (N=735) and the use of the validated "Signs of Inflammation in Children that Kill" (SICK) score provide a robust foundation for the authors' conclusions. The paper is written in excellent English and presents a compelling argument for HAP as a modifiable risk factor for pneumonia-related mortality.

B. Specific Recommendations for Improvement

1. Clarity on Clinical Assessment Timing (Lines 108–115)

The authors describe the calculation of the SICK score but do not explicitly state the timing of this assessment. While Table 1 suggests these were taken "at admission," the text should clarify if the SICK score represents a single baseline measurement upon hospital entry or a peak score during the stay. If it is an admission score, I recommend adding: "The SICK score was calculated upon initial hospital admission..." to line 111.

2. Magnitude of Association (Lines 182–187 & Figure 3A)

The authors report a statistically significant correlation between PM2.5 and the SICK score using Kendall’s tau (

τ=0.15. While this confirms a relationship, it does not provide a clinically intuitive sense of the effect size.

Recommendation: It would be highly valuable if the authors could provide a linear regression coefficient (β) to show the expected increase in SICK score points per unit (e.g., per 10 μg/m3) increase in PM2.5. This would allow clinicians to better understand the magnitude of risk.

3. Validity of the PM2.5 Modeling Approach (Lines 95–105)

The authors utilize a log-linear model to estimate exposure. While direct monitoring (e.g., personal samplers) is often preferred, the authors correctly argue that their quantitative modeling is more practical and "pragmatic" for a large-scale study in a resource-limited setting.

Recommendation: In the Discussion, the authors might briefly acknowledge that while the model has a moderate correlation (r=0.56) with measured values, it remains a superior alternative to simple dichotomous (Yes/No) fuel-type questionnaires, which are the standard in much of the existing literature.

4. Contextualizing Meta-Analysis Comparisons (Line 198)

The text mentions that the odds of pneumonia are 1.8 times higher in households exposed to solid fuels.

Recommendation: For clarity, please specify the comparison group used in this cited meta-analysis (e.g., compared to "clean-fuel households" or "non-exposed households").

5. Definition of the Multidimensional Poverty Index (MPI) (Table 1)

In Table 1, "MPI poor" is listed as a characteristic. The MPI is a complex composite index; the authors should briefly state the specific threshold or definition used to categorize a child as "MPI poor" in the footnote or Methods section to ensure reproducibility.

6. Formatting of Supplemental Material (Excel/CSV File)

Upon reviewing the supplemental data file, the data appears to be formatted as a raw CSV where all parameters are contained in a single comma-separated column (Column A).

Recommendation: To ensure the data is "Open Science" compliant and usable for other researchers, the authors should provide a properly formatted .xlsx or .csv file where each clinical and household variable is separated into its own clearly labeled column.

C.Conclusion

This study makes a strong contribution to the field by demonstrating that HAP is not just a risk factor for getting sick, but a driver of disease severity and mortality risk in children who are already hospitalized. The reasoning is sound, the limitations are honestly stated, and the clinical implications are clear. Following the minor clarifications above, the manuscript is well-suited for publication.

Reviewer #3: While reviewing the materials of the study, several questions and minor comments arose:

1. The study states that the average daily concentration of PM 2.5 is 145 μg per cubic metre. This is a rather dangerous concentration, as it is 10 times higher than the WHO standard. This is an important value for research, as it determines the prediction of risks to children's health. Therefore, it was necessary to clarify how this value was measured or calculated and to provide the calculation method or source for such a method.

2. The study should have listed the cities or towns in Uganda where the research was conducted. This does not violate the personal data of respondents but will allow for future analysis of data from the air pollution monitoring system on the streets of the studied areas to confirm that the health hazard to children comes specifically from the use of domestic stoves indoors.

3. It is known that stove heating and cooking are still used in many rural homes without centralised gas supply, including in Eastern European countries. However, carbon monoxide, which is produced by the incomplete combustion of organic fuel, is traditionally considered the main threat to residents there. In this regard, it was also worth describing the typical models of stoves used in Ugandan households.

6. PLOS authors have the option to publish the peer review history of their article (what does this mean?). If published, this will include your full peer review and any attached files.

Reviewer #1: No

Reviewer #2: **Yes:** Thomas C. Carmine MD

Reviewer #3: **Yes:** Buchavyi Yurii

---

## [Author Response · Author response to Decision Letter 1]

7 Apr 2026

30 March, 2025

Dear Editorial Team,

On behalf of my co-authors, thank you for the review of our revised manuscript (259525674.R1) entitled “Household air pollution is associated with disease severity in Ugandan children hospitalized with hypoxemic pneumonia.” The reviewer’s comments were most helpful and have increased the quality of our manuscript. We have addressed each comment in a point-by-point manner, below. We have submitted a revised manuscript, with and without changes highlighted.

Academic Editor comments:

1. The authors needs to provide the function from which the PM2.5 exposures were made from household characteristics along with its vital information about the model used.

Thank you. We have added the equation to the revised manuscript (reproduced below). We have included this in the Supplemental information in the revised submission:

E {log(〖PM〗_2.5)}=β_0+

β_F1 I (Fuel=Kerosene)+β_F3 I (Fuel=Wood or charcoal)+

β_K1 I (Kit=SOK)+β_K2 I (Kit=IDK)+

β_V1 I (Vent=Moderate)+β_V2 I (Vent=Poor)+

β_CH I (Cooking hours)

where I (X=L) = 1, if the categorical variable X assumes the level ‘L’, else 0.

Reference categories were: Liquified petroleum gas (LPG) for fuel; outdoor kitchen (ODK) for kitchen type/location; and “good” for ventilation.

Parameter a Coefficient Value

Intercept β_0 -1.653

Fuel b: kerosene vs. LPG β_F1 0.194

Fuel b: wood or charcoal vs. LPG β_F3 0.969

Kitchen: SOK vs. ODK β_K1 -0.389

Kitchen: IDK vs. ODK β_K2 -0.594

Ventilation: moderate vs. good β_V1 -0.082

Ventilation: poor vs. good β_V2 -0.391

Cooking hours β_CH 0.084

ODK, outdoor kitchen; IDK, indoor kitchen; SOK, separate outdoor kitchen; LPG, Liquified petroleum gas

a The region of India was included in the Balakrishnan model [9], but not here.

b Dung was not used as a fuel source by any study participants; therefore, its coefficient from the original Balakrishnan model [9] is not included here.

The daily average personal exposure of a young child was calculated from the kitchen area concentration as follows [10]:

〖PM〗_2.5 (daily average)= 0.628×〖PM〗_2.5 (kitchen)

2. Authors need to express the computational formula for the calculation of the sample size with input variable values. Only mentioning about the software is not appropriate.

Thank you. We have included the formula for calculating the sample size for a Pearson correlation coefficient in the Supporting information (reproduced below):

H_0:ρ≤0

H_a:ρ>0

n=((Z_(1-α)+Z_(1-β))/z_(r_a ) )^2+3

z_(r_a )=0.5×ln((1+r_a)/(1-r_a ))

Where:

ρ is the population correlation coefficient

r_a is the expected correlation

n is the desired sample size

Z represents the critical values for the alpha level and power

Z_(1-α)≈1.64 at α=0.05 (one-sided)

Z_(1-β)≈0.84 at β=0.2

3, It needs to be clarified why non-parametric test was used instead of parametric test for measuring comparative statistics.

Thank you. We used non-parametric statistics throughout (median and interquartile range for descriptive statistics, Mann-Whitney U-test for comparison of continuous variables in two groups, Kendall’s tau-B for correlations between continuous variables). This method avoids the assumption of normally distributed data (e.g., the PM2.5 is right-skewed). We have added this rationale in the methods section.

4. It needs to be clarified how the odds ratios of different outcomes for risk factors were calculated along with their confidence intervals in the methodology section.

Thank you. We have added the following details to the methods section:

The odds ratio (OR) and its 95% confidence interval (CI) were used to quantify the degree of association between binary variables. The OR was calculated as the cross-product of the 2×2 contingency table (OR=ad/bc) and the confidence interval was calculated using the maximum unconditional likelihood (Wald) method (normal approximation on the logarithmic scale) with the standard error calculated as 〖SE〗_(ln(OR))=√(1/a+1/b+1/c+1/d).

5. I could not find the questionnaire for the SICK score calculation with reference. Please add it in supplementary.

The following formula for the SICK score (with citation) was included in the Methods section:

The SICK score was calculated as previously described [12]. The score (range 0–8.6) was computed as the weighted sum of the following clinical variables: age < 1 month (+2.2) or < 12 months (+1.0) or < 5 years (+0.3); temperature > 38°C or < 36°C (+1.2); heart rate > 160 minutes−1 for infants or > 150 minutes−1 for children (+0.2); respiratory rate > 60 minutes−1 for infants or > 50 minutes−1 for children (+0.4); systolic blood pressure < 65 mmHg for infants or < 75 mmHg for children (+1.2); oxygen saturation < 90% (+1.4); capillary refill time > 3 seconds (+1.2); and level of consciousness less than “alert” (+2.0).

2.Thank you for stating the following financial disclosure: [This study was supported by Grand Challenges Canada (Grant Number 1909-27795 [MH]) and The Women and Children’s Health Research Institute (Reference Number WCHSSLDRP 2371).].

Thank you. We have included this Financial Disclosure statement at the end of the manuscript.

Thank you. We have included this statement at the end of the manuscript.

3. We note that there is identifying data in the Supporting Information file < Ahmed HAP data for analysis.csv>. Due to the inclusion of these potentially identifying data, we have removed this file from your file inventory. Prior to sharing human research participant data, authors should consult with an ethics committee to ensure data are shared in accordance with participant consent and all applicable local laws.

Please remove or anonymize all personal information (ID, AGE,), ensure that the data shared are in accordance with participant consent, and re-upload a fully anonymized data set. Please note that spreadsheet columns with personal information must be removed and not hidden as all hidden columns will appear in the published file.

Thank you. We have uploaded a revised file. Specifically, we have removed patient age and study ID number as these seemed to be the variables that were considered identifying.

Thank you. We have included Supporting information and captions in the main manuscript.

Not applicable.

There was one additional reference that needed to be added to the manuscript (to address a comment by Reviewer #2, see below): “Alkire S, Foster J. Counting and multidimensional poverty measurement. Journal of Public Economics. 2011;95(7-8):476-87.”

Reviewers' comments:

Reviewer #1: This manuscript presents relevant information regarding the exposure to household (HAP) - and not traffic-related - air pollution and the sickness severity among Ugandan children. It is very impressive because Uganda is one of the many low- and middle-income countries that rely on biomass combustion for cooking, heating and lighting. This manuscript presents data from a noteworthy large sample size, 735 children from 20 hospitals, and convincing results and a very clear discussion.

We thank the reviewer for these positive remarks.

However, I still have some doubts on how they estimated the PM2.5 levels.

Here are some minor points that should be addressed before accepting this manuscript for publication.

Minor comments:

# We need more details about the PM2.5 data. Was it measured in µg or estimated, like an index? Describe it better.

Thank you for the opportunity to clarify. We did not measure the PM2.5, but estimated it from a previously published equation, based on household and kitchen characteristics. The equation is a log-linear model linking the kitchen concentration of PM2.5 to household variables, based on past research [9]. In the revised manuscript we provided the equation in the Supporting information (reproduced below). This equation has been previously used for global burden of disease modeling [10], with correlation (r = 0.56) between predicted and measured values [9]. The model uses the following inputs to estimate the kitchen area PM2.5: fuel type, kitchen type, kitchen ventilation, and cooking duration. An additional variable from the original study, representing the state in India, was not included in our calculation. The ratio between the daily average personal exposure and kitchen concentration (0.628 for young children) was applied to estimate exposures for children in our study [10]. Household variables were collected using a standardized questionnaire, administered to the caregiver accompanying the hospitalized child, in the language best understood.

The following log-linear regression model [9] was used to estimate the PM2.5 in the kitchen area from household characteristics:

E {log(〖PM〗_2.5)}=β_0+

β_F1 I (Fuel=Kerosene)+β_F3 I (Fuel=Wood or charcoal)+

β_K1 I (Kit=SOK)+β_K2 I (Kit=IDK)+

β_V1 I (Vent=Moderate)+β_V2 I (Vent=Poor)+

β_CH I (Cooking hours)

where I (X=L) = 1, if the categorical variable X assumes the level ‘L’, else 0.

Reference categories were: Liquefied petroleum gas (LPG) for fuel; outdoor kitchen (ODK) for kitchen type/location; and “good” for ventilation.

Parameter a Coefficient Value

Intercept β_0 -1.653

Fuel b: kerosene vs. LPG β_F1 0.194

Fuel b: wood or charcoal vs. LPG β_F3 0.969

Kitchen: SOK vs. ODK β_K1 -0.389

Kitchen: IDK vs. ODK β_K2 -0.594

Ventilation: moderate vs. good β_V1 -0.082

Ventilation: poor vs. good β_V2 -0.391

Cooking hours β_CH 0.084

ODK, outdoor kitchen; IDK, indoor kitchen; SOK, separate outdoor kitchen; LPG, Liquified petroleum gas

a The region of India was included in the Balakrishnan model [9], but not here.

b Dung was not used as a fuel source by any study participants; therefore, its coefficient from the original Balakrishnan model [9] is not included here.

The daily average personal exposure of a young child was calculated from the kitchen area concentration as follows [10]:

〖PM〗_2.5 (daily average)= 0.628×〖PM〗_2.5 (kitchen)

-Lines 96-97: “We estimated the personal exposure of a young child to HAP using a log-linear model linking the kitchen concentration of PM2.5 to household variables [9].” Did you measure the PM2.5 concentration in the kitchens to estimate the real exposure? Mean measured PM2.5 should be presented in Table 2 too.

We did not measure the PM2.5, but estimated it from the previously published equation.

-Lines 101-103: “The ratio between the daily average personal exposure and kitchen concentration (0.628 for young children) was applied to estimate exposures for children in our study [10].” What exactly does this ratio mean? Explain better.

Thank you for the opportunity to clarify. The kitchen area concentration of PM2.5 is correlated with the daily average exposure. The daily average exposure varies between women, children, and men (0.742 for women, 0.628 for young children, and 0.450 for men). Thus, the daily average exposure for an individual can be estimated from the kitchen area concentration. These empirical values were determined from studies measuring 24-h kitchen and living area PM2.5 concentrations across 617 rural households drawn from 24 villages across four Indian states [9].

-“Statistically significant correlation between the PM2.5 and the SICK score (τ=0.15, p<0.0001). I cannot see this correlation in the graphic. What kind of variable was PM2.5 considered?

Figure 3 (Panel A) shows the data with trendline, and displays the rank correlation coefficient (Kendall’s tau-B, τ = 0.15, p<0.0001). PM2.5 was treated as a continuous variable (without assuming normal distribution). The correlation with SICK (also a continuous variable) was calculated using Kendall’s tau-B to avoid the assumption of normal distribution of data and to allow for tied ranks.

# Lines 282-284: “This estimate has a moderate correlation (r = 0.56) with measured values of PM2.5 [9]. This may have reduced the precision of the exposure assessment.” The R value of the correlation was already stated before, in the methods. So, you do not need to repeat it.

Thank you, we have removed this value from the Limitations paragraph.

Reviewer #2: A. General Comments

This is a well-researched, logical, and highly consistent manuscript that addresses a critical gap in global health literature. While there is extensive literature linking household air pollution (HAP) to the incidence of pneumonia, this study provides significant additional value by focusing specifically on the severity and prognosis of pneumonia among a high-risk, hospitalized, hypoxemic cohort.

We thank the reviewer for these positive remarks.

The large sample size (N=735) and the use of the validated "Signs of Inflammation in Children that Kill" (SICK) score provide a robust foundation for the authors' conclusions. The paper is written in excellent English and presents a compelling argument for HAP as a modifiable risk factor for pneumonia-related mortality.

Again, we thank the reviewer.

B. Specific Recommendations for Improvement

1. Clarity on Clinical Assessment Timing (Lines 108–115)

The authors describe the calculation of the SICK score but do not explicitly state the timing of this assessment. While Table 1 suggests these were taken "at admission," the text should clarify if the SICK score represents a single baseline measurement upon hospital entry or a peak score during the stay. If it is an admission score, I recommend adding: "The SICK score was calculated upon initial hospital admission..." to line 111.

Thank you for the opportunity to clarify and for the suggested revision. Indeed, we used data from the clinical assessment at hospital admission to compute the SICK score. We have added the following to line 112 of the revised manuscript: “The SICK score was calculated upon initial hospital admission…”

2. Magnitude of Association (Lines 182–187 & Figure 3A)

The authors report a statistically significant correlation between PM2.5 and the SICK score using Kendall’s tau (

τ=0.15. While this confirms a relationship, it does not provide a clinically intuitive sense of the effect size.

Recommendation: It would be highly valuable if the authors could provide a linear regression coefficient (β) to show the expected increase in SICK score points per unit (e.g., per 10 μg/m3) increase in PM2.5. This would allow clinicians to better understand the magnitude of risk.

Thanks for this valuable suggestion. We have utilized a linear regression analysis to find the coefficient (β) linking the log10(PM2.5) to the SICK score (range 0-8.6): β = 0.52 (95%CI 0.33-0.70 (p<0.0001). The reason for using the log-transformed PM2.5 is that the values are right-skewed and the measurement error in air quality data typically scales proportionally with the magnitude of the concentration. The coefficient in this model does have an intuitive interpretation: for every 10-fold change in the PM2.5, the SICK score increases by 0.52 units. We thank the reviewer for

---

## [Editor Report · Decision Letter 1]

14 Apr 2026

Household air pollution is associated with disease severity in Ugandan children hospitalized with hypoxemic pneumonia

PONE-D-25-36668R1

Dear Dr. Hawkes,

We’re pleased to inform you that your manuscript has been judged scientifically suitable for publication and will be formally accepted for publication once it meets all outstanding technical requirements.

Kind regards,

Srijan Lal Shrestha, Ph.D.

Academic Editor

PLOS One
---

## [Editor Report · Acceptance letter]

PONE-D-25-36668R1

PLOS One

Dear Dr. Hawkes,

I'm pleased to inform you that your manuscript has been deemed suitable for publication in PLOS One. Congratulations! Your manuscript is now being handed over to our production team.

Kind regards,

on behalf of

Dr. Srijan Lal Shrestha

Academic Editor

PLOS One